# The Effects of Light on Vertebrate Welfare: A Review

**DOI:** 10.3390/ani15223329

**Published:** 2025-11-19

**Authors:** Cristiano Schetini de Azevedo, Vinícius Donisete Lima Rodrigues Goulart, Cristiane Schilbach Pizzutto, Cynthia Fernandes Cipreste, Camila Palhares Teixeira, Robert John Young

**Affiliations:** 1Departamento de Biodiversidade, Evolução e Meio Ambiente, Instituto de Ciências Exatas e Biológicas, Universidade Federal de Ouro Preto, Campus Morro do Cruzeiro, Ouro Preto 35400-000, Brazil; cristiano.azevedo@ufop.edu.br; 2Transportation Research and Environmental Modelling Laboratory—TREM, Institute of Geosciences, Universidade Federal de Minas Gerais, Avenida Presidente Antônio Carlos, 6627, Belo Horizonte 31270-901, Brazil; viniciusdonisete@gmail.com; 3Laboratório de Ecologia e Evolução, Instituto Butantan, São Paulo 05585-000, Brazil; cspizzutto@yahoo.com.br; 4Belo Horizonte Zoo, Belo Horizonte 31365-450, Brazil; cycipreste@pbh.gov.br; 5Departamento de Ciências Biológicas, Universidade do Estado de Minas Gerais, Campus Ibirité, Avenida São Paulo (Rod. MG 049 URB), 3996, Vila do Rosário, Ibirité 32412-190, Brazil; camila.teixeira@uemg.br; 6School of Sciences, Engineering & Environment, Peel Building—Room G51, University of Salford, Manchester Room G51, Manchester M5 4WT, UK

**Keywords:** artificial light, circadian rhythm, zoo husbandry, visual perception, animal welfare, photoperiod, nocturnal behaviour

## Abstract

Visible light is essential for most animals because it controls daily and seasonal rhythms, influences behaviour, sleep, and reproduction, and even affects health and immunity. In the wild, animals are adapted to natural light cycles shaped by day and night, the moon, and the seasons. In captivity, however, animals are often exposed to artificial light that does not match these natural patterns. This can confuse their “internal clocks,” disturb sleep, alter activity, and reduce welfare. For example, animals kept in zoo nocturnal houses may experience reversed light cycles so visitors can see them active during the day, while zoo events at night can expose animals to prolonged illumination. Veterinary care can also involve continuous lighting that interferes with recovery. Although some artificial lights can help create safe and attractive environments for visitors, they may not provide the full spectrum of natural sunlight that animals need. New technologies, such as lamps that closely mimic natural light, offer promising alternatives, but their use is still limited. Understanding how different species respond to light and adjusting management accordingly is essential to ensure healthy and natural behaviours. By carefully managing light, institutions that house animals can enhance welfare outcomes and, in the case of zoos and similar facilities, promote education and public engagement.

## 1. Introduction

Not only modern zoos, but all institutions that house animals, prioritise maintaining high levels of welfare for the individuals in their care [1,2,3]. In addition to the ethical issue [4], high levels of welfare improve animal health [5,6], improve the relationships with other individuals and with humans [7,8], may increase reproductive rates [9,10], and increase the quality of life of animals [5,11]. Improving animal welfare can be based on the five domains model [12,13], where positive experiences are enabled in the four physical welfare domains (i.e., health, environment, nutrition, and behaviour and their interactions), positively influencing the fifth domain (i.e., mental) [14,15].

Considering the environmental domain, most captive animals must be offered complex and stimulating environments [16], as this provides them with a greater chance of interacting with their environment and behaving more naturally, exhibiting a greater diversity of positive behaviours [17]. Complex environments, if designed considering the biology of the species kept within them, improve animal welfare [18,19,20]. However, one aspect of this environmental complexity that is still little studied in terms of its influence on the welfare of captive animals is light. Despite this being a topic of considerable interest for human well-being and, recently, for wildlife [21,22,23].

Light can be defined as an electromagnetic wave capable of exciting the cells of the visual system of animals; that is, it is a natural agent that makes things visible by stimulating vision [24,25]. It is important to note that “visible light” is species-specific; that is different species can perceive different light waves (e.g., colours) [26,27]. Light can influence animals’ circadian and hormonal rhythms [23,28,29], sleep and activity patterns [30,31,32], can be used in medicinal therapies [33,34], and facilitate the visualisation of animals in zoos with night houses or that promote nocturnal visits [35,36]. The stress that altered light regimes can generate in animals, such as behavioural anomalies (e.g., sleep, space use, foraging, and migratory behaviours) [30,32,37,38] and conditions such as sleep disorders and seasonal affective disorders (SADs), however, is little studied in the context of zoos compared to the ecological context, especially concerning artificial lighting generated by humanity [30,39,40]. Therefore, it is important to consider the impact of light on the well-being of captive animals.

In this review, we will examine various aspects of light, including how vertebrates perceive it, its impact on the biology of species, and how modern zoos can utilise this knowledge to improve welfare by also considering the lighting regimes to which their animals are exposed. Future research on the topic, as well as a decision tree for animal managers, will be highlighted, as there are still considerable gaps in our knowledge of this subject.

## 2. The Physical Characteristics of Light and Their Relevance to Animal Welfare

Light is an electromagnetic wave that propagates rapidly without requiring a medium (speed of light in vacuum: 3.0 × 10^5^ km/s) [41]. As an electromagnetic wave, light is characterised by its wavelength, frequency, and intensity, as well as its ability to be reflected, refracted, or diffracted [42,43]. The wavelength defines the distance between successive wave crests and determines the colour perceived by the visual system. The frequency, measured in Hertz, represents the number of oscillations per second and is inversely proportional to the wavelength, where higher frequencies correspond to shorter wavelengths and greater photon energy [44,45,46]. Terrestrial animals are primarily exposed to sunlight, though additional sources such as the moon, fire, bioluminescence, and artificial lighting can also influence their physiology and behaviour. The visible portion of the light spectrum (approximately 380–750 nm) is perceived as colours, but sensitivity to ultraviolet or infrared light varies widely among taxa [26,27]. This variation has important implications for zoo environments, as the spectral composition of artificial light can alter how animals perceive colours, detect food items, or recognise conspecifics. For instance, if lighting does not match the visual sensitivity of a given species, it may distort visual cues, reduce foraging efficiency, and impair social or reproductive behaviours [47,48,49,50,51].

Beyond wavelength and intensity, the way light interacts with surfaces (through reflection and refraction) has direct consequences for animal perception in captivity. Reflected light determines how objects, conspecifics, and enclosure boundaries appear to the animal [48,49]. Smooth surfaces produce specular reflections that may generate sharp, mirror-like images, while rough textures create diffuse reflections that scatter light in multiple directions, allowing animals to perceive shapes and colours from various angles [44]. These optical properties influence essential behaviours such as navigation, foraging, and visual communication [50,51,52,53]. In aquatic or semi-aquatic species, refraction at the air–water interface can alter depth perception and the apparent position of prey or objects [54,55,56,57], demanding behavioural and cognitive adjustments that may be affected by enclosure design. Therefore, understanding the physical properties of light is essential for developing lighting systems and exhibit structures that support natural visual experiences, reduce stress, and promote the welfare of animals under human care.

Due to the spherical shape of the Earth and its axial tilt (approximately 23.5°), the incidence of sunlight varies substantially according to latitude [58]. Regions close to the equator receive more direct solar radiation throughout the year, resulting in greater light intensity and a relatively constant day length, with around 12 h of daylight and 12 h of darkness [59]. In contrast, higher latitudes, both north and south, experience marked seasonal variations. During the summer, these regions receive sunlight for prolonged periods, reaching up to 24 h of daylight in polar areas, while in winter, the days become extremely short, with very little or no sunlight (for example, the polar night phenomenon) [60]. This variation results in marked differences in the amount of light available throughout the year, which poses challenges for visual perception in natural environments. Consequently, various animals have evolved specific visual adaptations to cope with these variable conditions, adjusting aspects such as light sensitivity and visual acuity to maximise the capture and processing of ambient light [61]. The structures and mechanisms responsible for light perception in animals will be explored in the next section.

## 3. Light Perception in Vertebrate Animals: Eye Structure and Vision Evolution

The vertebrate eye has a fundamental structural organisation that includes the cornea, lens, iris, pupil, retina, sclera, and choroid, which together ensure the capture and processing of light for vision [62] (Figure 1). The cornea acts as the main refractive element, the lens fine-tunes image focus, and the iris controls pupil size to regulate the quantity of light entering, while partial reflection arises from ocular surfaces such as the retina (and, in some species, the *Tapetum lucidum*) [63]. The retina, containing rods (light intensity-perceiving cells) and cones (colour-perceiving cells), converts light into neural signals relayed via the optic nerve, while the sclera provides protection and the choroid supplies vascular support [63]. Diffraction may occur at the pupil aperture but has little functional relevance for vision [63]. This basic plan varies among vertebrate groups to meet different ecological and functional demands. For instance, birds possess a highly vascularised *Pecten oculi*, which is not only thought to enhance retinal nutrition and visual performance but is also hypothesised to function as a photoreceptive apparatus, mediating daily or seasonal periodic behaviour via non-visual opsins [64,65]. These non-visual opsins are light-sensitive proteins distinct from those used for image-forming vision; instead, they function to detect light levels and photoperiod, thereby regulating physiological processes and behaviours such as circadian rhythms and seasonal breeding cycles [66,67]. In some fish species, the cornea is structurally modified or functionally divided to allow vision both in air and underwater [62,68]. Such modifications exemplify the adaptive radiation of the vertebrate eye in response to diverse visual environments [63,69,70].

Many aspects of animal life are intrinsically linked to the activity periods of individual groups or species. Depending on the specific activity period, features such as eye size, corneal diameter, and light sensitivity may vary among different animal groups. Consequently, two primary activity patterns can be defined: diurnal or photopic, referring to activity in high-light environments, and nocturnal or scotopic, active after sunset in conditions of limited luminosity [71]. Some animals may be crepuscular, active solely at dawn and dusk, or cathemeral, exhibiting a propensity for activity at any time throughout the day [72]. Prior studies suggest the existence of a common organisational principle that links activity patterns to ocular morphology [73].

The structure and functionality of eyes vary across animal groups, being adapted to the prevailing conditions of the natural environments in which each group evolved, including factors such as light incidence, darkness, and the ambient light spectrum. Nocturnal birds typically possess a relatively larger corneal diameter in proportion to the axial length of the eye, a characteristic that enhances light sensitivity; conversely, diurnal birds exhibit eyes with a greater axial length relative to the cornea, thereby prioritising visual acuity [73]. For both nocturnal and diurnal reef fish, light intensity demonstrably influences eye size and morphology [74]. The correlation between eye size and morphology in lizards and their visual perception across varying light conditions has also been highlighted; for example, the eyes of nocturnal lizards, like those of many nocturnal terrestrial vertebrates, are adapted to low-light environments, featuring widely dilating pupils to maximise light capture from each object [75]. In aquatic mammals, ocular anatomy exhibits numerous adaptations for both underwater and aerial vision, encompassing alterations in eye shape and size, alongside modifications to the structure of the retina and optic nerve [76].

In mammals, corneal size reflects both the activity pattern and the luminosity conditions of the habitat, with notable adaptations evident in diurnal and cathemeral species [77]. Animals inhabiting forested environments possess proportionally larger corneas than those found in open habitats, indicating an adaptation to low light conditions [78]. However, among nocturnal animals, no variation was observed between habitats, suggesting an intense selective pressure to maximise light capture, irrespective of the specific environment [78].

For animals living under human care, simulating natural conditions regarding light types, twilight, and darkness is essential for their physical and psychological well-being, as inadequate artificial lighting can lead to vision problems and/or behavioural stress. In vertebrate vision, photoreceptors (rods and cones) line the posterior part of the eye. Rods are primarily associated with low-light vision, whilst cones gradually become the predominant light-perceiving cells as light levels increase [79].

The ability of visual systems to resolve rapid changes in light, known as temporal resolution, is particularly important when animals are exposed to artificial lighting. A standard measure of this capacity is the critical flicker fusion frequency (CFF), which represents the highest flicker rate at which a light is still perceived as flashing rather than continuous [80]. Species with elevated CFFs can detect faster light fluctuations and are thus more prone to perceiving flicker from artificial light sources. When the flicker frequency of a lamp approaches or falls below an animal’s CFF, the resulting visible flicker may induce visual discomfort, stress responses, and alterations in behaviour [81,82].

In managed environments such as aquaria, zoos, or farms, it is essential to consider the CFF of the species being housed. Flickering lights, if below the temporal resolution of the animal, may interfere with prey detection, navigation, social interactions, and overall welfare. While modern high-frequency LED drivers can mitigate this risk (operating at >1000 Hz), older lighting technologies (e.g., standard fluorescent lamps with mains frequency flicker, <100 Hz) may produce frequencies that are perceptible to many vertebrates and invertebrates and therefore may still induce visual discomfort or stress. Table 1 provides approximate CFF ranges for major taxa and common artificial lighting frequencies to guide the assessment of potential visual stress.

Vision constitutes a crucial perceptual stimulus, enabling organisms to communicate and acquire information. As one of the most versatile and vital sensory modalities, it allows organisms to perceive and respond to their surroundings with precision. The interplay between light, specialised ocular structures, and neural processing transforms physical stimuli into meaningful perceptions [83]. These interactions, coupled with environmental diversity, have led to the evolution of a diverse range of visual systems, each adapted to the specific characteristics of its respective environment.

Considering the diversity of environments in which animal groups have evolved, it is essential to be attentive to the differences and capacities of their visual systems. The range of wavelengths visible to other animals often diverges significantly from that of humans, owing to the presence of different visual pigments in cones that absorb in different regions of the spectrum [84]. Despite the significant challenge of light scarcity and obscuring neural noise (where the brain’s inherent electrical activity masks weak visual signals) in low-light environments such as moonless nights or the deep sea, research over the past fifteen years has remarkably demonstrated that nocturnal and deep-sea animals, even those as small as 1mm, exhibit formidable visual abilities essential for their orientation, navigation, and environmental interaction [85]. These adaptations, therefore, offer important insights into the evolution of vertebrate vision.

### The Evolution of Visual Systems in Vertebrates

Vision in vertebrates is highly diversified, reflecting specific ecological and behavioural adaptations. Different groups, including fish, amphibians, reptiles, birds, and mammals, exhibit significant variations in ocular structures, light sensitivity, and perceptual abilities, such as ultraviolet (UV) perception and monochromatic vision [62].

Driven by gene duplications and environmental pressures on light capture, the evolution of vertebrate vision has resulted in the diversification of opsin proteins and other retinal components into pigments sensitive to varied spectral ranges, a molecular diversity which, combined with morphological variations in the eye and unique neural pathways, accounts for the distinct different processing of light and colour across fish, amphibians, reptiles, birds, and mammals [86,87,88,89].

In vertebrates, the diversity of visual systems reflects adaptations to their varied environments, as well as to the incidence of light. Generally, the evolution of vertebrate eyes involves the expansion and specialisation of opsin families (SWS, RH2, LWS, RH1; opsins are light-sensitive G protein-coupled receptors located in the photoreceptor cells of the retina; [90]), in addition to alterations in the rod-to-cone ratio and modifications to intraocular filters. Examples include lipid droplets (oil droplets) in birds and reptiles, and the lens and *Tapetum lucidum* in mammals, which collectively refine both spectral sensitivity and light intensity sensitivity [91,92]. These biochemical and morphological changes maximise sensitivity in low-light environments through the predominance of rods and molecular adaptations in RH1. Specifically, rods are photoreceptor cells highly sensitive to dim light, enabling vision under scotopic (low-light) conditions, while RH1 (rhodopsin) is the opsin found in rods, optimised for absorbing faint light [93]. This prevalence of rods and the enhanced efficiency of RH1 facilitate the detection of even very few photons, thereby increasing overall light sensitivity [94]. However, as rods are generally monochromatic, relying primarily on this single type of photoreceptor reduces the ability to distinguish between different wavelengths of light, thus leading to a diminished chromatic discrimination [95,96]. Conversely, in high-light environments, a greater proportion of cones leads to elevated spectral discriminability, which is highly beneficial for identifying food, attracting mating partners, and facilitating social signals [91,92].

Fish exhibit remarkable genomic and ontogenetic plasticity, as many teleosts retain a duplicated opsin repertoire, enabling them to regulate gene expression based on depth, ontogeny, and environmental spectrum [97]. Deep-sea species frequently display ontogenetic progression: cone dominance in the larval stage, transitioning to rod dominance in adulthood; in extreme cases, they may possess multiple functional copies of RH1 to capture low-intensity light [98,99].

In amphibians, the evolution of opsins has resulted in significant diversity, including the presence of non-visual opsins with adaptive functions beyond vision [88,89]. Light-sensitive opsin genes exhibit functions beyond vision, encompassing the regulation of circadian rhythms and seasonal responses [100]. In anurans, a diversity of non-visual opsins is observed, reflecting evolutionary adaptations to luminosity and environmental characteristics [88].

Diurnal reptiles, for the most part, exhibit ocular adaptations that modulate light transmission, as they preserve trichromacy and are sensitive to UV light. The functional role of this vision includes prey detection, sexual signalling, and behavioural thermoregulation. Variation between species is influenced by lifestyle habits and local spectral availability [86,101]. As an example, four types of lipid droplets (oil droplets) were identified in the retina of lizards from the family Lacertidae: two pigmented and two colourless [102]. One of these droplets transmits ultraviolet radiation, suggesting an association with UV-sensitive cones. The combination of optical, molecular, and behavioural data demonstrated that these lizards possess functional ultraviolet vision, supported by a system of four cone types, including one specific for UV [102].

Birds exhibit special adaptations in their visual structures, including high sensory sensitivity and the ability to recognise constellations for orientation and navigation. Furthermore, the size and weight of the eyeball can vary significantly, from about 8 mm in kiwis to 50 mm in ostriches; in some owls, the eyes can account for up to 32% of the head’s weight, compared to only ~1% in humans [103]. The avian eye is unique among anatomical structures due to its characteristics specifically adapted to interpret light intensity (brightness) and wavelength (colour). Species inhabiting darker environments, such as undergrowth and tropical regions, and possessing long-range vision, tend to have larger eyes better adapted to capture dim light and maintain visual acuity [104]. Some researchers investigated the optics of photoreceptor cones in chickens (*Gallus domesticus*) and found that the pigmented oil droplets in the cones primarily function as spectral filters, rather than light accumulators, by adjusting the spectral sensitivity of the cones through filtering specific light ranges [105]. Furthermore, the mitochondrial ellipsoid aids in efficient light transmission by improving the optical coupling between the cone’s inner segment and the lipid droplets [105].

In mammals, the evolutionary history of a “nocturnal bottleneck” has led to a partial loss of cone diversity in many clades and to anatomical and molecular adaptations favouring vision at low light levels, such as the *Tapetum lucidum*, accompanied by a higher proportion of rods [87,98]. Also, factors such as eye size and adaptation to light influence visual acuity and behaviour in diverse environments [106]. The mammalian visual system is highly diverse, reflecting adaptations to varying light conditions. Rods are more abundant in nocturnal species, whereas cones predominate in diurnal species [107,108]. Many nocturnal mammals show a reduction in the number of cones, especially short-wavelength sensitive cones (S-cones); for instance, nocturnal primates like *Aotus trivirgatus* have completely lost this type of cone [83]. Conversely, rodents such as the fat-tailed gerbil (*Pachyuromys duprasi*) and the house mouse (*Mus musculus*) utilise S-cones to detect ultraviolet light, which aids behaviours such as predator avoidance and resource localisation during twilight hours [83].

Vision in black and white, or monochromacy, arises from the absence of cone types necessary for colour discrimination [109]. In mammals, this condition can manifest in two ways: rod monochromacy, where all cones are absent, or cone monochromacy, where only one cone type, typically the L-cone, is retained, preventing colour perception [110]. Rod monochromacy is characterised by low visual acuity and a complete lack of colour discrimination in low-light conditions, in addition to daytime blindness or hemeralopia [111]. Dolphins, whales, and seals have lost the SWS1 gene, which is responsible for the perception of short wavelengths (blue/violet), retaining only long-wavelength cones (L-cones) and rods, resulting in cone monochromacy [92].

Like other sensory structures that enable organisms to interact with their environment and other living beings, eyes have evolved to perform specific functions in capturing and processing light [62]. This evolution occurs only when modifications increase the organism’s fitness, meaning they result in more effective behaviours based on visual information [112].

Nilsson [109] proposes four increasing levels of visual capacity:Non-directional photoreception: Detects light versus darkness.Directional detection: Identifies the direction of light.Low-resolution vision: Capable of detecting crude shapes or movements.High-resolution vision: Allows for the perception of detailed images.

According to this author, each advance in the eye emerged in response to more demanding behavioural needs. The evolution of eyes thus followed a functional sequence driven by increasingly complex behaviours, starting from simple light detection, followed by directional detection and rudimentary vision, culminating in high-resolution vision, which is present in more advanced vertebrates, cephalopods, and arthropods [112]. However, this process did not occur along a single evolutionary line: similar types of photoreceptors and ocular architectures arose multiple times independently in distant lineages, as a result of typical functional constraints rather than solely direct evolutionary kinship [112,113].

The formation of the vertebrate eye represents a significant leap in how individuals acquire information from and communicate with their environment. The common ancestor of lampreys and gnathostomes, living approximately 500 million years ago, already possessed a well-developed chambered eye, similar in many aspects to that of modern vertebrates [91]. This suggests that the basic structure of the vertebrate eye emerged very early in evolutionary history. One of the main events in this evolutionary process was the differentiation of photoreceptors [114]. The evolution of opsins and complex layered retinas conferred unique flexibility upon the vertebrate visual system, a capability further enhanced by the retina’s duplex organisation (comprising both cones and rods), which facilitates practical ocular function across a broad spectrum of light intensities [91].

Research into the early evolution of vertebrate photoreception, notably through studies of lampreys and lungfish, reveals distinct evolutionary stages in their visual systems: lampreys exhibit five photoreceptor types with specific opsins, indicative of photopic vision and potential colour perception, whereas lungfish possess a more complex retina featuring multiple cones and rods, suggesting a more developed colour vision and adaptation to diverse wavelengths [115]. This diversity in retinal morphology and opsin repertoires indeed reflects differing evolutionary paths in the development of the vertebrate visual system. Furthermore, the mention of critical genes like PAX6 and RAX underscores the profound connection between developmental processes and the evolution of the eye [91].

Having explored the evolutionary journey and diverse adaptations of vision across various animal taxa, it becomes evident that these highly refined sensory systems are not simply passive light receptors. Instead, they represent fundamental conduits through which environmental light plays a critical role in shaping an animal’s behaviour, physiological processes, and overall welfare.

## 4. How Light Can Influence Animal Welfare

Recent research has demonstrated how artificial light in the human world can cause a whole series of physical and psychological health problems for humans [116,117]. Despite this research, many places keeping animals use human-appropriate lighting regimes [118], such as zoos, farms, pet shops, and individual owners. Light constitutes a fundamental environmental factor that modulates animal behaviour, physiology, and welfare levels [40]. Its influence extends from regulating daily behavioural cycles to seasonal and reproductive responses, mediated by complex neurobiological and hormonal mechanisms [119,120]. Understanding these processes through the lens of the five domains of welfare (nutrition, health, environment, behaviour, and mental state) [12,13,14] is of considerable importance, particularly for effective captive management and mitigating the impacts of increasing artificial light pollution.

### 4.1. Nutrition

Light exposure can indirectly affect nutrition by altering circadian rhythms, which regulate physiological processes such as foraging, food intake, energy metabolism, and nutrient assimilation [121,122,123]. Seasonal changes in photoperiod influence metabolic processes, appetite, and nutrient assimilation. For example, reduced photoperiods may suppress foraging activity, leading to altered energy intake [124,125]. Intrinsically, photosensitive retinal ganglion cells (ipRGCs) play a pivotal role, detecting light and transmitting signals to the suprachiasmatic nucleus (SCN) in the hypothalamus, the central pacemaker of the biological clock [40,126,127]. Exposure to light at specific times can ‘reset’ the circadian clock, a process known as entrainment (the process by which circadian rhythms are synchronised with environmental cues) and influence patterns of locomotor activity, foraging, and sleep [128,129]. Understanding these patterns is essential to optimise feeding schedules and support nutritional welfare in captivity.

The disruption of natural light cycles, particularly through artificial light at night (ALAN), can have significant effects. In horseshoe bats (*Rhinolophus pusillus)*, for instance, ALAN delays their emergence from roosts, leading them to miss the period of peak prey availability. This phenomenon directly impacts their foraging opportunities and, consequently, their food intake [130]. Furthermore, ALAN can mask natural light cues essential for synchronising animals’ circadian clocks, thereby altering daily activity patterns. It also increases the perceived risk of predation for nocturnal species, which can inhibit foraging-related activities [131].

Seasonal photoperiod changes notably affect metabolic processes, appetite, and nutrient assimilation, with varied responses across species. In broiler chickens, the photoperiod directly influences feed consumption and efficiency. A 20-h light photoperiod has been shown to result in the highest growth rate across all ages, while improved feed efficiency correlates with shorter day lengths, suggesting that extended periods of darkness may reduce metabolic maintenance requirements [132]. In dairy cows, light duration affects dry matter intake, and for calves, extended photoperiods (18h Light:6h Dark) increase colostrum intake and average daily gain [133]. Similarly, weaned piglets exposed to 23 h of light exhibit increased feed intake and energy metabolism rates, with an extended photoperiod of 8 to 16 h improving daily weight gain [133]. These physiological and behavioural responses are frequently mediated by hormones, such as melatonin, which regulates the secretion of prolactin and other crucial hormones involved in growth, maturation, and reproduction [133].

Conversely, for some species, increased ambient light, including artificial sources, can enhance foraging opportunities. Common redshank (*Tringa totanus*), for example, benefits from artificial illumination originating from industrial complexes at night. This improved visibility facilitates more efficient sight-based foraging, replacing their usual tactile methods. Birds in these well-lit areas consequently spend a greater proportion of time foraging, indicating a preference for visual foraging and an extension of nocturnal feeding opportunities under such conditions [51]. Moreover, for fish larvae, a minimal light intensity is essential for effective prey detection, capture, and ingestion, where the synergistic effect of food availability and photoperiod is a significant determinant of growth [134]. Longer photoperiods or continuous daylight can stimulate growth in numerous fish species, including salmonids, primarily through increased food intake and, potentially, superior food conversion rates [134].

Additionally, it is important to consider that light with an incomplete or altered spectrum, commonly emitted by artificial lighting, may impair food selection in species with a broader visible range. Many birds, for instance, perceive ultraviolet (UV) and near-ultraviolet wavelengths that are absent from most artificial light sources [135,136]. Such spectral deficiencies can modify how fruits or other food items are visually perceived, influencing their apparent ripeness or nutritional value based on variations in sugar composition or pigmentation [137]. Experimental evidence demonstrates that spectral cues directly affect foraging decisions. For example, zebra finches (*Taeniopygia guttata*) show altered preferences for coloured seeds when specific wavelength bands are removed, indicating that access to the full visual spectrum is critical for natural food selection [138]. Likewise, blue tits (*Cyanistes caeruleus*) rely on UV cues when detecting cryptic insect prey, and their foraging efficiency decreases when ultraviolet wavelengths are filtered out, demonstrating that UV perception enhances their ability to locate food under natural conditions [139]. Similar mechanisms may operate in fishes, as visual spectral sensitivity strongly influences prey detection, preference, colour discrimination, and feeding efficiency under different light spectra [140,141]. Consequently, animals may experience reduced agency in food selection and a diminished perception of their environment when housed under lighting conditions that lack key spectral components. Although the welfare implications of these spectral constraints are less severe than those associated with circadian disruption, they still represent a relevant dimension of environmental quality and behavioural opportunity in captivity.

A comprehensive understanding of these complex light-mediated patterns is therefore essential for optimising feeding schedules and ensuring the nutritional welfare of vertebrates in captive environments. This necessitates carefully adapting light conditions to align precisely with the specific biological needs and physiological stages of each species [133].

### 4.2. Health

Light exposure plays a fundamental role in regulating an animal’s health, influencing essential physiological processes such as hormonal balance, immune responses, and sleep–wake cycles. Light exerts significant control over hormonal secretion, particularly melatonin [142]. Light exposure inhibits melatonin production by the pineal gland, directly impacting sleep–wake cycles and reproductive timing in mammals [143,144]. Beyond melatonin, circadian rhythms regulate hormones such as cortisol and insulin, which are essential for metabolism and stress responses [145]. Seasonal changes in light also influence the immune system, with reduced photoperiods being associated with suppressed immune responses [146,147], the endocrine system, through altered secretion of hormones such as melatonin and cortisol, regulates reproductive and stress physiology [148,149]; and metabolic function, by shifting energy balance and feeding behaviour in response to photoperiodic cues [31,150] (Figure 2).

Ensuring animal welfare in captivity should comprise a 24/7 approach, which demands continuous and dedicated attention, with particular emphasis on the quality of sleep and rest [151]. Furthermore, health issues, such as degenerative diseases, can compromise the ability for adequate rest, thereby highlighting the importance of proactive management and veterinary attention [152]. For most vertebrates (except some cave-dwelling species or species that live permanently underground), adequate exposure to natural daylight and darkness at night is essential for the hormonal regulation governing sleep–wake cycles, directly impacting animals’ ability to attain the necessary rest for their physical and mental well-being [153]. Technological advancements, such as infrared cameras, accelerometers, and data loggers, have facilitated the monitoring of these resting patterns, providing fundamental tools for the continuous assessment of animal welfare [154,155]. Moreover, devices designed to record inactivity yield valuable insights into the amount/quality of rest or sleep achieved, and understanding the activity–rest patterns of species under human care is crucial for detecting alterations that may indicate whether the light regime to which animals are exposed is appropriate or requires adjustment [156,157].

The relationship between sleep, light, and welfare is significantly influenced by ALAN and its effects on circadian rhythms. Studies indicate that sleep disruptions can lead to increased diurnal inactivity and abnormal behaviours, as observed, for example, in chimpanzees, which are sensitive to nocturnal disturbances [158]. Giant pandas, conversely, demonstrate that when housed outside their natural light conditions, they exhibit reduced activity levels, indicating that altered light exposure affects their circadian and circannual rhythms [159]. ALAN can be considered light pollution and has been demonstrated to negatively affect the behaviour and biological rhythms of nocturnal species, such as the grey mouse lemur (*Microcebus murinus*), which exhibited desynchronisation of its daily activities and potential long-term adaptation problems when exposed to this condition [160]. Furthermore, studies have shown alterations in the development and body size of amphibians, as well as the induction of oxidative stress at all life stages of this group, with significant implications for their behaviour, physiology, and overall welfare in light-polluted environments [161].

Given that ALAN interferes with melatonin production, which is also an important hormone for regulating reproductive cycles in mammals, it also has negative impacts on reproductive processes [162]. Examples have been documented, for instance, for tammar wallabies (*Notamacropus eugenii*) exposed to urban light pollution, which experienced delayed births due to masked seasonal light cues, potentially affecting offspring survival by misaligning them with resource availability [163]. The reproductive timing of birds, such as European blackbirds (*Turdus merula*), has also been altered, with reproductive systems developing up to a month earlier when exposed to artificial night light, thereby misaligning them with environmental conditions [164]. Another example has been observed with great tits (*Parus major*), as their egg-laying dates advanced, particularly under white and green light, with effects varying according to spring temperature [165]. By obscuring the detection of short days and inappropriately signalling long days, ALAN disrupts seasonal reproductive activities, leading to decreased individual fitness and negatively impacting overall ecosystem dynamics [166]. Consequently, changes in reproductive timing can lead to mismatches in offspring survival rates and parental investment, affecting population stability and biodiversity [163].

Finally, seasonal changes in light exposure can lead to Seasonal Affective Disorders (SADs), characterised by mood and behavioural alterations [129]. Light, through its influence on the circadian pacemaker and dopaminergic activity, is fundamental for regulating seasonal behaviours [167]. SAD in zoo animals, particularly in diurnal species, has attracted increasing attention due to its implications for welfare and management. Research with the diurnal African grass rat (*Arvicanthis niloticus*) indicates that phototherapy can effectively mitigate SAD symptoms. Animals exposed to dim light, for instance, demonstrate increased depression-like behaviours compared to those in bright light conditions [168]. Bright light therapy has been shown to reverse alterations in locomotor activity and dopaminergic signalling in these animals, suggesting a promising therapeutic approach for SAD [129].

The mechanisms of light action in mood and behaviour regulation are complex. The orexinergic system, involved in sleep regulation and reward pathways, may play a crucial role in how light exposure influences these aspects in diurnal rodents [39]. Furthermore, studies indicate that light exposure can affect gene expression related to circadian rhythms, establishing a direct link between phototherapy and mood regulation [129]. In the management of captive animals, phototherapy can improve the welfare of diurnal species, especially during winter months when natural light is limited. This approach aligns with findings on the efficacy of phototherapy in treating symptoms of SAD [167]. However, variability in responses across species and the need to account for individual variability within a species highlight the complexity of SAD management. This underscores the need for essential additional research to optimise these interventions across diverse animal populations.

In addition to the effects on SAD, ultraviolet (UV) light is essential for the health of reptiles and amphibians, influencing their behaviour, physiology, and general welfare [169]. Adequate UV exposure is vital for metabolic and reproductive processes, while a deficiency can lead to diseases. Conversely, ALAN can also induce oxidative stress in these animals [170], underscoring the importance of a careful balance in managing the light environment in artificial settings. Standardised lighting protocols adapted to the needs of each species can significantly improve the welfare and health of animals kept under human care, and the natural light regimes of the species’ native habitats may serve as valuable reference points for reproducing appropriate conditions in captivity [118].

### 4.3. Environment

Light is a pervasive component of the environmental domain, shaping the conditions under which animals live and thrive. While predominantly provided by solar radiation during the day, light is also a significant presence at night, originating from natural sources such as the moon, and increasingly, from artificial illumination. Within this environmental context, the lunar cycle also exerts a significant influence on animal behaviour, primarily mediated by variations in light levels and the synchronisation of biological rhythms [171]. These effects are observed across diverse species, which exhibit complex adaptive mechanisms in response to lunar cues. Lunar illumination alters ambient light levels, modulating hormonal activities such as the production of melatonin and GnRH, which in turn affect reproductive cycles and other behaviours [172]. Furthermore, genetic and molecular mechanisms, such as the L-cryptochrome gene, enable animals to distinguish lunar phases, regulating their circalunar clocks [172,173].

Behavioural adaptations to moonlight include lunar phobia (avoidance of light) and lunar philia (attraction to light), which are strategies related to predation risks and foraging opportunities [174]. Many nocturnal mammals, particularly rodents, exhibit lunar phobia, reducing their activity during full moon nights to avoid predators, and it is estimated that 16% of tropical rainforest mammal species exhibit lunar light phobia, compared to only 3% that show philia [174]. Complementarily, data show that 30% of terrestrial mammal species living in these forests avoid the full moon, in contrast to 20% of species that exhibit attraction to it [174]. In this regard, the authors draw attention to the influence of moonlight on animal behaviour despite the forest canopy, signalling a significant impact for degraded and fragmented forests. This critical dependence on lunar cycles for natural behaviours raises a considerable concern for captive environments, where artificial lighting often creates conditions akin to a perpetual full moon, potentially disrupting these finely tuned adaptations.

In birds, species such as the common nighthawk (*Phalaenoptilus nuttallii*) and the whip-poor-will (*Caprimulgus vociferus*) utilise moonlight for foraging, synchronising nesting cycles [175,176]. Eastern whip-poor-wills may even avoid migration during the full moon, using the light for pre-migratory fattening [177]. Nocturnal primates rely on moonlight for foraging [176], while on full moon nights, felines such as ocelots may reduce their movement distances to minimise risk and visibility; conversely, lynx select denser vegetative cover to optimise hunting [178].

Amphibians, such as the Eastern grey treefrog (*Hyla versicolor*), adjust their reproductive activities based on lunar phases to synchronise breeding and reduce predation [179,180]. Tungara frogs (*Engystomops pustulosus*) adjust partner preferences according to light levels [176]. Marbled salamander (*Ambystoma opacum*) larvae may emerge in response to decreasing illumination during eclipses, linked to prey availability [180].

Marine animals, such as corals and annelid worms, exhibit circalunar reproductive rhythms, depending on both endogenous clocks and external lunar cues [173]. Although well-documented, the extent and nature of lunar influence can vary significantly. Factors such as artificial light pollution and habitat degradation may alter these natural behaviours, requiring further research for a complete understanding of these dynamics.

### 4.4. Behaviour

Light regimes are fundamental drivers of animal behaviour, influencing activity patterns, social dynamics, and the expression of species-specific traits that are essential to welfare. Light directly influences activity patterns, reproductive behaviours, and social interactions [156,181]. Circadian entrainment synchronises locomotor activity, foraging, sleep, and reproductive cycles [166,182,183]. Artificial Light at Night (ALAN) can disrupt these patterns, leading to abnormal behaviours or decreased activity in nocturnal and diurnal species alike [166,184]. Furthermore, lunar cues modulate foraging, predator avoidance, and reproductive timing across diverse taxa, from amphibians to mammals and birds [172,175,178,180]. Additional examples of light-mediated behavioural responses across different species and contexts are discussed throughout other sections of this review.

In domestic animals, the management of light parameters, such as photoperiod, intensity, and spectrum, significantly affects various behaviours. For pigs, light influences activity patterns, with higher activity observed during lit hours, and a preference for resting in dimmer conditions versus elimination in brighter ones [185]. Dim conditions are also associated with more negative social interactions, while brighter light leads to more positive interactions and improved social discrimination in growing-finishing pigs [185]. Photoperiod adjustments in pigs can also reduce abnormal behaviours such as inactive standing and over-exploring [133]. Similarly, poultry exhibit refined behavioural responses to light spectrums, with laying hens showing a preference for natural-like light over standard artificial light, often resulting in increased active behaviours such as locomotion, standing, and foraging [186]. Intermittent lighting in broilers, while potentially improving growth, may elevate stress levels and negatively impact welfare, highlighting the need for sufficient dark periods for rest [133].

Beyond domestic species, ALAN poses significant threats to wild nocturnal mammals. The endangered Stephens’ kangaroo rat (*Dipodomys stephensi*), for instance, reduces its foraging activity in the presence of artificial light, with impacts extending far from the light source and mediated by lunar conditions. These rodents actively avoid artificially lit patches, likely due to increased visibility to predators or temporary blindness caused by their highly light-sensitive eyes [38]. In zoo environments, the nocturnal activity and sleep patterns of large mammals like giraffes are strongly shaped by age and the presence of darkness. Giraffes display a polyphasic sleep pattern with rest-activity cycles of approximately 4 h during the dark phase, during which activity generally increases with age, and juveniles spend more time in REM sleep [187].

For institutions managing animals and allowing visitor presence, such as zoos, management must consider visitor interactions during night visits or inside night-houses, including the use of flashlights or crowd-induced light disturbances, which can amplify behavioural stress. Implementing dimming schedules, red-light covers, or technology-driven lunar cycle simulations can mitigate these impacts, allowing animals to express species-typical behaviours and promoting more natural interactions with their environment and conspecifics [188,189].

### 4.5. Mental State

As a fundamental environmental factor, light’s influence extends across an animal’s four functional domains (environment, nutrition, health, and behaviour), thereby generating a spectrum of experiences, both positive and negative, that critically shape its ultimate mental welfare. Consequently, mental well-being is intrinsically linked to light, as it can disrupt sleep, circadian rhythms, and animal behaviour.

Specifically, the physiological impacts of light are directly linked to affective states. For instance, UVB radiation, present in natural light and activatable by artificial sources, can promote the cutaneous synthesis of vitamin D3 [190]. UVA/UVB exposure has been shown to influence the expression of genes regulating circadian rhythms and the production of beta-endorphin, a hormone associated with feelings of relaxation and well-being in pigs [185]. Conversely, environments deficient in ultraviolet (UV) wavelengths can elevate basal corticosterone levels and increase fear and stress in species like poultry [186]. The disruption of natural light cycles forces nocturnal animals like the Stephens’ kangaroo rat to alter essential behaviours such as foraging, which, by increasing perceived predation risk, can induce chronic stress and negatively impact mental well-being [38]. Furthermore, the quality and duration of sleep, directly influenced by light-dark cycles, are paramount for mental restoration and brain development, as evidenced by polyphasic sleep patterns and age- and darkness-dependent variations in REM sleep duration in giraffes [187].

Additionally, abrupt and unpredictable changes in illumination can elicit rapid behavioural and physiological reactions that may have direct welfare consequences. Sudden light–dark or dark–light transitions reliably evoke startle- and escape-type responses in fishes (including larval and adult zebrafish, *Danio rerio*), with large, rapid increases in locomotor activity that can produce uncontrolled acceleration and increase the likelihood of collisions with tank walls or viewing windows [191]. Such abrupt transitions have also been used experimentally as unpredictable stressors, producing anxiogenic-like responses and altered behaviour in vertebrate models, which supports the idea that intermittent, unexpected lighting can be a welfare-relevant stressor in captive animals [192]. Experimental evidence also demonstrates that light wavelength and intensity can affect growth, immune function, reproductive physiology, and behavioural activity in fishes. For example, blue LED light enhances growth performance, antioxidant capacity, innate immunity, and oocyte development in sweetfish (*Plecoglossus altivelis*), whereas red light can inhibit feeding motivation and induce oxidative stress [193]. Similarly, in cultivated juvenile Pacific bluefin tunas (*Thunnus orientalis*), sudden changes from darkness to 300 lx induced high swimming speeds and momentary bursts of activity before complete retinal adaptation, particularly in fish around 40 days after hatching, leading to collisions with tank or net walls and contributing to elevated mortality in this age class [194]. These findings indicate that even in natural or semi-natural environments without predators, abrupt changes in light intensity can alter behaviour and physiology in fishes. Although practitioners and institutional reports (e.g., anecdotal observations in aquaria and zoos) suggest that sudden changes in lighting may sometimes result in injuries or heightened stress in captive animals, we found no peer-reviewed studies directly documenting this phenomenon. Nevertheless, given the reported effects of abrupt illumination on behaviour and physiology, it appears plausible that sudden light changes could negatively affect welfare in managed systems, representing an important area for future empirical research.

Providing animals with predictable, controllable lighting conditions and enriched environments supports positive mental states. Moreover, aligning light exposure with natural rhythms, including lunar and seasonal cycles, enhances overall welfare by fostering psychological comfort and stability [195,196]. The captive environment of zoos is used as an illustrative example in the following section, as it provides a well-studied context where the effects of lighting on animal welfare can be observed and managed in a controlled yet complex setting.

## 5. How Zoos Manage Light

Zoos house a wide range of vertebrate taxa, each possessing distinct sensory abilities and highly specialised ecological adaptations for perceiving light. Despite this vast variation in eye structure and visual perception across species, lighting regimes in many captive environments are often primarily configured for human convenience and to satisfy stringent health and safety regulations (ex., UK Statutory Instruments 1992, No. 3004, Regulation 8 about light in the workspace) [197]. This human-centric approach, however, frequently results in a ‘one size fits all’ lighting standard that demonstrably fails to meet the highly diverse and specific photobiological requirements of the various vertebrate species housed within these institutions. Consequently, the artificial light in a zoo frequently fails to reflect the nuanced conditions of natural lighting pertinent to each species [198]. A striking example of this mismatch is found in the display of nocturnal mammals; an estimated 70% of all mammalian species are primarily nocturnal, yet many are exhibited to the public during daylight hours or under constant daylight-like conditions in their enclosures [199]. In the context of artificial light pollution, this diversity makes light a highly variable husbandry factor. Consequently, there is no universal solution to mitigating the welfare impacts of artificial light across taxa in zoos. To address this complexity, institutions should systematically integrate operational requirements with species-specific welfare needs, enabling the development of targeted strategies that reduce the effects of light pollution on animals in captivity. Thus, housing animals with distinct habits and behaviours requires technical-scientific knowledge regarding the species’ natural environment, their activity peaks, and their general behaviour.

### 5.1. How Are Animals Exposed to Artificial Light in Zoos?

Zoos usually receive visitors during the day; however, in groups such as mammals, almost 70% of species are nocturnal [199]. To allow visitors to experience nocturnal animals, some zoos house nocturnal animals in artificially reversed light cycles, where the environment is kept dark during the day and artificially illuminated during the night. This management practice presents challenges related to the variation in light cycles throughout the year (i.e., seasons) and the type of light technology employed. In a review survey for lorisid primates in North America, only 13% of zoos calculate the day lengths based on the latitude and mimic the twilight periods, and most animals were exposed to some artificial light during the 24 h of the day [200,201]. While the use of natural, full-spectrum light would be ideal for promoting plant growth and creating a visually enriched environment, its application in nocturnal exhibits is challenging due to the need for light–dark cycle inversion. Nevertheless, specific artificial lighting systems now provide near full-spectrum illumination [202]. However, despite claims of their low cost [202], a comprehensive evaluation of their broader feasibility for widespread use in nocturnal exhibits, critically including their comparative economic implications against conventional lighting systems, is still needed to optimise animal welfare and exhibit quality [203].

A challenge in allowing visitors to a reversed light cycle exhibit is preventing the use of photography flashes or even the widely available mobile phone flashlights, as some people might turn the flashlight on to move around in a dimly lit environment [203]. Visitors in such exhibits should be aware of the dim environment and be mindful of any visual disability before the visit. The gradual change in light made by the exhibit design order would allow visitors to adjust their sight and avoid turning on the flashlight to move around [203]. Furthermore, the strategic placement of ground-level lighting can effectively guide visitors through these dimly lit environments, ensuring safe navigation without the need for personal flashlights and thereby maintaining the integrity of the exhibit for the nocturnal animals. Practical examples from zoological institutions illustrate how light management can support animal welfare in nocturnal species. For instance, the use of red-spectrum lighting and reversed light cycles in nocturnal exhibits housing slow lorises (*Nycticebus* spp.) has been shown to maintain species-typical activity patterns and reduce stress-related behaviours [204]. Similarly, controlled experiments on pygmy slow lorises demonstrated that dim red light conditions promoted natural foraging and locomotor activity, improving environmental engagement compared to brighter or blue light environments [205]. Bats have been extensively studied with respect to the behavioural and physiological impacts of artificial light, and a range of mitigation strategies, such as the use of warm-spectrum lighting, directional shielding, and preservation of dark corridors, are already well established [206]. These findings underscore how thoughtful lighting design, such as the use of low-intensity, warm-spectrum lighting and gradual light transitions, can enhance animal welfare while maintaining visitor experience.

Night events in zoos present a valuable opportunity for promoting conservation and awareness in the public [207]. For instance, over 50 North American AZA-accredited zoos participate in celebrations involving holiday light displays and evening activities [208]. However, artificial illumination may exacerbate the impacts of nocturnal public visitation and those stemming from prolonged exposure to artificial light [209,210]. One of the most recognisable effects is the change in activity cycle, with diurnal species being the most negatively affected [211]. Exposure to artificial light and crowds during nighttime events can have adverse effects on animals, and care should be taken when planning such events [207]. When hosting events in zoos, managers should be aware of mitigation strategies for reducing the additive impacts of light, noise, and extended visitation on animals [212]. Behavioural and hormonal measures confirm the impacts from hosting events and adding new features to the captive environment [213]. When properly mitigated, zoos can minimise their impact on sensitive animals [208].

During nighttime, animals in zoos might be exposed to sources of artificial light that are not from the enclosure itself, but from operational areas, such as pathways, security, parking, and others. Those might be overlooked, but they might present a significant source of artificial light pollution. While animals in natural environments can often avoid artificially lit areas, those under human care are unable to escape these confined, illuminated environments. As it cannot be avoided, in some cases, even low levels of artificial light will affect exposed species [131]. Studies on Artificial Light at Night (ALAN) advocate for the efficient use of lighting, limiting its use where necessary and adjusting its timing, intensity, and spectrum [184]. Nevertheless, an increasing awareness highlights the necessity for innovative lighting strategies that balance public engagement with animal welfare. The Singapore Night Safari offers a compelling case in point, having implemented a unique illumination system designed to emulate lunar conditions within its enclosures, exemplified by the Pangolin Trail [214]. This approach enables visitor observation while carefully mitigating the environmental impact on nocturnal species. Such systems, often involving advanced LED technology to produce light with specific spectral qualities and intensities, align with broader recommendations for ecologically sensitive lighting. These recommendations typically involve reducing short-wavelength (blue and UV) emissions, favouring warmer colour temperatures (e.g., below 2700K), employing minimal light levels, and ensuring precise directional control, with activation restricted to essential periods, to minimise disturbance to wildlife, including species such as bats [215]. It might sound contradictory, but the cheaper LED illumination allows the indiscriminate use of light, increasing the exposure to blue wavelengths to which many biological responses are particularly sensitive [195]. One factor that might create a conflict of interest between security staff and animal welfare professionals is that the use of higher levels of luminance is related to a higher perception of security, which could reflect higher levels of light pollution [216]. ALAN can be used indiscriminately also for aesthetics (e.g., ornamental light and architecture) and advertisement (e.g., billboards); in both cases, the lack of mitigation measures can contribute to the light exposure of animals [195].

Animals under veterinary care, such as in intensive care settings, can be exposed to continuous lighting as it is required to support professionals in their activities. Animals in this setting can experience sleep disturbances similar to those reported in human Intensive Care Unit (ICU) patients [217]. Prolonged disruption of circadian rhythms through artificial illumination may impair immune function and delay recovery, as has been demonstrated in both experimental animal models and clinical studies [218,219]. Furthermore, inappropriate light regimes in veterinary hospital annexes or recovery wards can exacerbate stress and hinder healing, underlining the importance of designing lighting schedules that support physiological rest–activity cycles [118,220].

### 5.2. What Is the Source of Artificial Light?

The use of different light sources in zoos presents unique challenges for animal welfare. Incandescent, fluorescent, and LED lamps each have distinct spectral properties, and none reproduce the full spectrum of natural sunlight. Incandescent lights, while once common, are inefficient and emit high levels of infrared but little ultraviolet radiation, while compact fluorescent lights offer greater efficiency but have discontinuous spectra and can produce high-frequency noise perceptible to some animals [221,222]. Fluorescent light can even create a secondary type of impact on animals, which would be noise. Some fluorescent lights produce continuous, high-frequency sounds that could be imperceptible to humans but potentially loud/stressful for animals with hearing in this range [198,223]. LEDs, now widely adopted, are highly energy efficient, yet typically emit blue-rich light designed for human vision, which may disrupt circadian rhythms and lack ultraviolet wavelengths critical for specific taxa [224,225]. These technological differences underscore the importance of selecting and managing artificial light systems in accordance with species-specific sensory and physiological needs. The types of light available have undergone significant changes in recent years, with a trend of continued growth expected in the future, and it is gaining worldwide attention [226].

For customers, LEDs can be purchased by colour temperature, measured in Kelvin, which indicates cooler and warmer lights. This measure is not an emission measure but is commonly used to describe light properties from a human-centric perspective. The light properties during the day and night are different, and the use of technological approaches to simulate natural conditions, such as daylight and moonlight, can be achieved by emulating the wavelength (light spectrum), angle, and duration [203]. One important consideration regarding the development of technologies and improvements in energy efficiency and costs is the potential for increased illumination in previously unlit areas [227]. The increased waste and spill of artificial light could potentially affect species due to the unplanned use of illumination.

In contrast to human perception, artificial environments often expose animals to lighting conditions that are both unfamiliar and disruptive. Artificial illumination can produce light and dark areas with sharp contrasts, which may be a significant perceptual threat for animals but an imperceptible difference for humans [228]. Exposure to unnaturally bright illumination, constant light without natural variation, abrupt on/off transitions, and spectral gaps compared to natural full-spectrum light can all interfere with circadian rhythms and physiological processes [166,229,230]. For many species, such aberrant lighting represents a significant source of environmental stress, with the potential to alter behaviour and reduce overall welfare in captivity [198].

The reduction of light intensity is a husbandry practice to reduce aggression in production animals; however, it is detrimental to welfare [198]. On the other hand, bright environments can induce fear responses and induce fights [231,232]. Reptiles and amphibians are often kept in environments with low access to natural light (due to the need for heating). UV lighting exposure is essential for the synthesis of vitamin D, and the use of the right artificial illumination technology is required for creating an appropriate microhabitat for these species [233]. For primates, the use of the proper exposure to ultraviolet B lamps in indoor enclosures has been applied successfully and safely for preventing metabolic bone disease, outperforming dietary supplementation [234]. However, less attention is given to providing artificial UV-B to nocturnal species, even though in the wild they may be exposed to this as they rest in the daylight [203].

Light in natural environments contains a full, continuous spectrum, ranging from ultraviolet through visible to infrared, with intensity and spectral balance changing across the day and seasons. In contrast, artificial light sources, such as LEDs, rarely provide the full spectrum [235]. Highly efficient LEDs, while optimised for human vision with their blue-rich emission (450–495 nm), present significant sensory constraints for animals as their lack of UV wavelengths and strong short-wavelength bias can disrupt circadian entrainment, thereby distorting or omitting crucial spectral cues to which their visual systems and physiological processes are attuned [236]. The flickering of fluorescent lights has a significant effect on birds, with responses detected both in behaviour and corticosterone levels, demonstrating how an event imperceptible for humans can negatively impact animal welfare [222].

The different periods of the day, with higher light incidence, low luminosity, or darkness, as well as the varied types of environments, have shaped throughout evolution how animals behave and perceive the world around them. Eye size and ocular structure play a fundamental role in the habits of these animals, determining, for example, how they communicate through specific colours for mate selection, how they forage, where they hunt, and how they orient themselves in low-light environments [73]. These characteristics also confer adaptive advantages, such as the capacity for ultraviolet or monochromatic vision. Understanding the habits and physical characteristics of animals is essential for practical conservation actions and habitat protection, ensuring that each species can thrive and fulfil its role in healthy environments.

### 5.3. Animal Welfare in the “Light” of the Five Domains

In controlled environments, such as zoos and aquariums, maintaining diverse species from distinct groups is always a challenge, as the management of these animals involves actions on multiple fronts within the five domains of welfare. Providing positive experiences in nutrition, health, environment, behavioural interactions, and mental state requires attention, sensitivity, knowledge, and technical-scientific expertise [13,14,188,237]. For instance, the light regime within an enclosure significantly impacts the physical environment and subsequently influences an animal’s health and mental domain. Enclosures must meet the needs of animals as a species and as unique individuals, to stimulate the exhibition of normal and natural behaviours, which are among the leading indicators of good welfare levels [189,238].

Furthermore, as previously mentioned, effective light management for animals, particularly those housed in nocturnal exhibits, is also contingent upon managing public behaviour. The impact of artificial illumination, often emanating from visitor devices such as mobile phone flashlights, can have a synergistic effect with noise disturbances, thereby exacerbating potential stressors. Consequently, zoological institutions must proactively engage with their visitors to encourage appropriate conduct and minimise disturbance around the animals. Such public-animal interactions can therefore be analysed through the lens of the Five Domains, directly influencing behavioural interactions, altering the environmental domain via public-generated light and noise, and potentially compromising the animals’ mental state. To mitigate these potential negative impacts and foster a positive visitor experience, zoos and aquariums consequently invest in activities designed to educate and enhance public engagement.

Zoos and aquariums have a broad public reach, granting them a unique opportunity to promote engagement with essential topics such as biodiversity conservation, planetary health, human well-being, and the promotion of a sustainable lifestyle [1]. To this end, these institutions conduct activities that bring visitors closer to animals through chats, games, and recreational activities, which result in engagement through recreation. Some institutions offer guided night visits, with a restricted number of visitors, during which nocturnal animals can be observed in their enclosures. Conducting these visits requires care and attention to ensure they are not stressful, both for diurnal and nocturnal animals. The itinerary should include only nocturnal animals, avoiding routes near the enclosures of diurnal animals; even so, silence must be maintained. For nocturnal animals that receive visits during twilight or night periods, it is important to use the minimum amount of light possible, so as not to disturb or confuse the animals. To ensure comfort for the animals and visibility for visitors, flashlights used to aid viewing can be covered with red cellophane paper, reducing the incidence of white light and thereby providing a positive experience for the visitor while contributing to the animals’ welfare.

During the full moon period, nocturnal visits can be conducted using minimal artificial lighting, which can significantly contribute to the welfare of the visited animals. This is particularly relevant given that, in nature, animals often remain hidden during a full moon due to increased vulnerability from the brightness generated by reflected sunlight. However, animals under human care, free from the threat of predators, may experience greater ease in moving around their enclosures during a full moon, even if their behaviours are like those exhibited in the wild. This principle could also be applied to animals housed indoors for extended periods, where creating artificial lunar cycles of light could mimic natural conditions and support their well-being, such as for various species during winter or in indoor enclosures. A hardware–software system originally designed to reproduce lunar cycles and simulate moonlight illuminance under controlled laboratory conditions has been developed, with potential applications in zoological settings [239]. Moreover, providing environmental enrichment stimuli can positively encourage animals to display their natural behaviours during these nocturnal visits. These nocturnal educational experiences offer the public a unique opportunity to observe active nocturnal animals, fostering a deeper connection with nature and facilitating important attitudinal changes towards conservation. To further enhance these experiences and reduce reliance on intrusive artificial light, zoos might consider providing visitors with readily accessible technology such as night vision goggles, binoculars, and cameras, which are now increasingly affordable and could significantly enrich the observation of nocturnal species. An example of this is the devices provided for guests during night safaris at Disney’s Animal Kingdom Lodge’s Starlight Safari [240]. Indeed, educational initiatives involve the transmission of knowledge generated from studies and conservation projects, in addition to stimulating and encouraging attitudinal changes that can lead to positive impacts on the preservation of natural environments and species [241]. Therefore, management and welfare activities are key factors in motivating visitors to learn about animals and their world, both in zoos and aquariums, as well as in nature.

Welfare levels are directly linked to good management practices, which should promote positive experiences [242]. The enclosure should offer choices that allow the animal to control its surroundings and interactions with the physical environment and other group members, enabling it to respond adequately to stimuli. For instance, providing adequate shelters in quantities compatible with the number of animals, spots for sunbathing and others with shade and rain protection, retreat areas, private spaces, foraging opportunities, and resting areas are essential measures for the comfort and welfare of animals [243,244]. In this sense, light functions as a physical-sensory stimulus, underpinning appropriate and healthy animal-environment interactions by allowing for the strategic deployment of heating lamps, coloured lamps, and other elements that temporarily modify the environment, thereby offering varied sensations and enabling animals to choose distinct areas within the enclosure [245]. However, the extent to which animals should directly control ambient lighting, such as having on/off switches for primary light sources, warrants careful consideration. While large prey species may appear to benefit from constant illumination because of perceived safety, such exposure to Artificial Light at Night (ALAN) can inadvertently compromise welfare by disrupting natural photoperiods, modifying melatonin production, suppressing nocturnal behaviours, intensifying light pollution, and potentially contributing to Seasonal Affective Disorders (SADs) [246,247,248,249]. Therefore, rather than direct control over foundational lighting, the focus should shift towards implementing sophisticated, welfare-optimised lighting strategies that are precisely managed to support natural rhythms, offer appropriate environmental enrichment, and allow for choice in specific, controlled microenvironments, without compromising overall physiological and behavioural health [250].

To this end, a proactive approach involves deploying carefully designed lighting solutions that prioritise animal well-being. For example, advanced systems exist that precisely regulate light exposure to support natural physiological processes, such as specialised stable lighting engineered to enhance equine health and condition [251]. Concurrently, whilst acknowledging that artificial lighting is often essential for human health and safety within enclosures, as in ensuring keeper visibility and facilitating maintenance, responsible implementation is paramount. This includes strategies such as utilising localised, floor-level pathway lighting for staff, rather than broad ceiling illumination that can disrupt animal photoperiods and create conditions akin to a perpetual full moon. Moreover, mitigation techniques developed for bats in response to Artificial Light at Night (ALAN), such as adopting warm-spectrum lighting (<2700 K), directing light downward with shields or baffles, establishing dark buffer zones, and limiting unnecessary illumination, could serve as valuable baselines for designing zoo lighting schemes that minimise disruption and enhance welfare across a broader range of species [206]. In sum, the thoughtful application of varied light types and intensities is fundamental; it can shape the environment into a source of optimal welfare or, conversely, a source of stress, should planning fail to align adequately with the physical, physiological, and behavioural characteristics of the species in question.

The incidence of light must accommodate the specific behaviour and habits of animals, such as diurnal, nocturnal, crepuscular, or cathemeral, as well as the environment in which the species evolved. In this regard, maintaining the quality of the environment in relation to the type of light and its intensity is essential. A study with captive felids, for example, revealed that while cheetahs maintained a 24-h activity rhythm similar to wild individuals, lions adapted their behaviour, exhibiting increased daytime and twilight activity compared to their wild counterparts, with both species displaying activity peaks at dawn and dusk, often preceded by increased movement near feeding times, underscoring natural light as a vital temporal marker dictating patterns of rest and activity [252].

Despite the relatively well-understood effects of light on animals in the scientific literature, its specific implications for the welfare of zoo and aquarium animals, particularly in relation to the Five Domains of welfare, remain an understudied area. This highlights significant research gaps that need to be addressed, especially regarding appropriate light management strategies to meet the high welfare standards expected of these institutions.

## 6. Future Studies: The Necessity of More Studies Evaluating the Light Impact on Captive Vertebrates and How to Mitigate Them

In summary, light is an environmental variable that governs multiple dimensions of animal life, from circadian rhythms and seasonal cycles to foraging, reproduction, behaviour, and predator-prey interactions [164,166,253,254]. In nature, natural light regimes are structured by daily photoperiod, lunar cycles, and seasonal variation [255]. Artificial light at night (ALAN) disrupts the natural light-dark patterns to which animals have adapted over evolutionary time, potentially rendering them susceptible to detrimental effects when exposed to artificial regimes, such as those frequently present in zoological environments [163,256]. Such alterations can compromise welfare by disrupting sleep–wake patterns, altering hormonal regulation, and increasing stress or abnormal behaviours [257,258]. Despite the growing recognition of light pollution as a global ecological factor, this form of pollution remains largely overlooked, and systematic evaluations of its impact in zoos and other captive settings are still significantly under-researched. Notably, ALAN has been associated with increased obesity in human populations and laboratory rodents [259,260,261]. Given the reported obesity among some zoo animals, particularly, it is pertinent to question whether ALAN might be a contributing factor.

A study conducted at Denver Zoo during an event where lights operated every night throughout December and January of 2017/2018, attracting numerous visitors, can illustrate both the potential and limitations of current approaches to welfare assessment in zoological environments [208]. Whilst the results did not reveal significant increases in aggression in great Indian hornbills (*Buceros bicornis*) during exposure to artificial light and noise, they underscore the necessity for physiological analyses that complement behavioural data to more directly quantify stress levels. Another interesting study involved a multi-species assessment of a Christmas lights event at Knowsley Safari (UK) on how different taxa respond variably to anthropogenic stressors, such as artificial light, noise, and increased visitor presence [262]. Although the study found no negative welfare impacts, the observed behavioural responses, such as increased retreat behaviour in capybaras and heightened vigilance in tapirs, demonstrate that animals can cope with events in species- or context-specific ways, at least in the short-term. Importantly, a significant challenge in these evaluations is that physiological or behavioural responses could be chronic and species-specific, taking years for adverse consequences, such as reduced fertility or longevity, to become evident [247,263,264].

Studies in zoological settings, to date, have neglected a more integrated perspective that simultaneously considers behaviour, physiology, and welfare. Moreover, most studies on the consequences of light pollution in captive animals either extrapolate from wild populations or focus on single-species case studies [265]. This gap hinders a comprehensive understanding of how artificial light may impact the long-term health and well-being of captive animals. Thus, it becomes necessary to develop standardised and evidence-based protocols that integrate both behavioural and physiological indicators, allowing for the assessment of animal welfare during after-hours events in zoos [266,267]. Furthermore, these assessments must be species-specific and include long-term post-event monitoring to effectively identify any delayed or cumulative effects. Recent research indicates that only 7.6% (n = 310) of articles on zoo welfare include direct assessments of captive conditions, and few studies have implemented standardised protocols for assessing light exposure [268], which hinders the establishment of inter-institutional comparisons and evidence-based guidelines. These research gaps are particularly critical, as zoos are characterised by their roles as conservation centres and educational institutions [2]. Research needs to move beyond single-point measurements or short-term observations by adopting more integrated methodologies, as without such integration, assessments risk underestimating the cumulative or delayed impacts of artificial lighting on animal welfare.

New technologies can serve as powerful tools to enhance welfare assessments. Automated video tracking systems, acoustic sensors, and machine-learning-based behaviour recognition have been increasingly utilised to quantify animal activity and behavioural states with high precision [269,270]. The incorporation of these innovative approaches into zoo welfare research can enable systematic, replicable, and multimodal assessments of how artificial lighting biologically influences animals.

The necessity of defining species-specific light requirements stems from the varied visual systems among vertebrates, where nocturnal mammals possess rod-dominated retinas for higher sensitivity to dim light, while many birds and reptiles are tetrachromatic or sensitive to UV radiation, thus justifying distinct luminosity needs based on their unique photoreceptor combinations and neural circuits [271,272]. This diversity demonstrates that a single lighting standard is unlikely to be suitable for all species housed in zoos. It is fundamental that future research experimentally explores different wavelengths, light intensities, and photoperiod regimes in representative groups. Thus, it will be possible to identify limits that minimise the adverse effects of artificial lighting, while simultaneously ensuring adequate visibility conditions for both animal management and environmental education initiatives.

Minimising exposure to disruptive wavelengths is paramount for mitigating the effects of light on captive animals. Research suggests that lighting management interventions, such as adapting habitats using artificial illumination that simulates solar and lunar cycles, can aid in maintaining the circadian cycle and promote natural behaviours [27,203]. Another interesting form of mitigation is spectral mitigation, which advocates for the use of long-wavelength light sources (≥560 nm; amber/red), given that evidence suggests red light, for instance, in aye-ayes significantly increased melatonin levels compared to exposure to blue light [35], reinforcing the implication of wavelength as an essential factor related to the welfare of captive animals.

A novel, and perhaps the most advanced, alternative would involve the construction of lighting devices that can be controlled by the animals themselves, allowing them to choose between different lighting regimes and types. This approach aligns with the recurring discussion on providing individuals with choice and control over their environment, tailored to their specific needs and preferences [273,274]. Human–animal interaction mediated through technology, particularly computerised devices designed for animals, can promote greater autonomy, enhance social interaction, and provide cognitive challenges that stimulate their mental abilities [275]. Thus, an animal’s ability to activate light by choice could serve as an effective stress-reduction strategy, allowing them to solve problems and express preferences, thereby significantly contributing to their welfare through increased autonomy in environmental management. Such measures offer practical alternatives for aligning lighting regimes in zoos with the sensory needs of species, consequently mitigating health and welfare risks.

Zoological institutions should integrate key practices into their routine operations, including undertaking species-specific assessments of the impact of artificial light on each individual animal, gathering diverse data to establish a robust scientific basis for management protocols, and consistently offering environmental education to visitors. These comprehensive efforts are essential for a thorough understanding of animal welfare, particularly within the Five Domains framework [13,14,188,237]. To support future research on this topic and guide professionals working in zoos and other animal-holding institutions in the comprehensive assessment and mitigation of Artificial Light at Night (ALAN) impacts on animal welfare, a systematic decision tree is proposed (Figure 3).

The framework commences with an Initial Assessment (Step 1) to determine whether a species is currently exposed to ALAN or if a new/modified lighting regime is planned. Positive identification in either scenario necessitates progression through the subsequent steps. Characterisation of Species and ALAN Environment (Step 2) involves a detailed contextual analysis. This step requires determining the species’ activity pattern (e.g., diurnal, nocturnal), identifying species-specific biological requirements (e.g., visual system characteristics, photoperiod adaptations, UV-B needs), and meticulously mapping ALAN sources and their parameters (duration, intensity, spectrum, spatial distribution). Following characterisation, an Establishment of Baseline and Multi-modal Welfare Assessment (Step 3) is conducted. This involves collecting both behavioural (e.g., activity levels, sleep–wake patterns, social interactions) and physiological data (e.g., hormone levels, body condition) during ALAN exposure, benchmarked against a pre-ALAN or control baseline. An integrated analysis correlates ALAN parameters with welfare indicators to detect immediate significant changes. Crucially, this step also addresses potential chronic effects, advocating for long-term monitoring to identify delayed or cumulative adverse impacts before proceeding. Based on identified negative impacts, Targeted Mitigation Strategies (Step 4) are developed and implemented. These multi-faceted strategies encompass species-specific lighting designs (tailored for diurnal, nocturnal, crepuscular, or cathemeral animals), general lighting control principles (spectral, intensity, and temporal mitigation), spatial adjustments (e.g., shielded lighting, unlit retreat areas), enhancement of animal choice and control over their light environment, and human-centred adjustments (e.g., non-intrusive visitor observation tools). The final stage, Evaluation and Refinement of Mitigation Effectiveness (Step 5), involves re-assessing animal welfare using the same multi-modal protocols as in Step 3. The aim is to confirm whether welfare indicators have improved or stabilised under the new lighting regime. If desired outcomes are not achieved, an iterative process of re-evaluation and adjustment of mitigation strategies is initiated, ensuring an adaptive and effective ALAN management approach.

## Figures and Tables

**Figure 1 animals-15-03329-f001:**
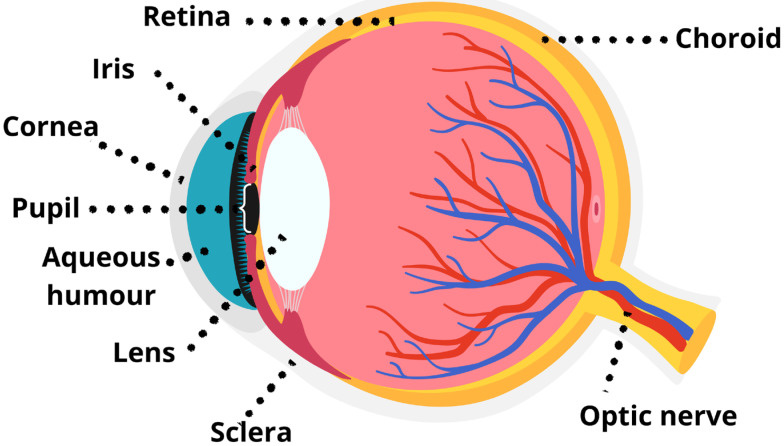
Fundamental structural organisation of the vertebrate eye. This diagram illustrates the principal anatomical components of the vertebrate eye, encompassing the cornea, lens, iris, pupil, retina, sclera, optic nerve, aqueous humour, and choroid. These structures collectively facilitate the precise capture and processing of incident light, thereby enabling visual perception (Authors’ own diagram).

**Figure 2 animals-15-03329-f002:**
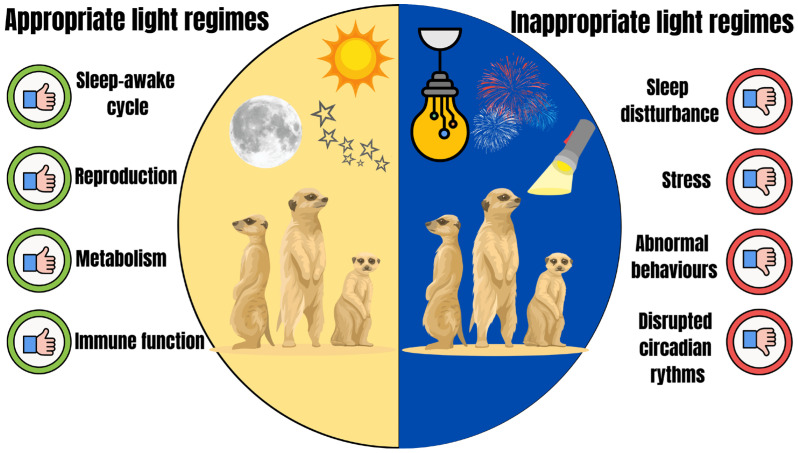
Conceptual diagram showing the influence of light regimes on animal biology and welfare. Both natural and artificial light can regulate circadian rhythms, sleep, reproduction, and health, but inappropriate regimes may cause disruption, highlighting the importance of management tailored to species’ needs (Authors’ own diagram).

**Figure 3 animals-15-03329-f003:**
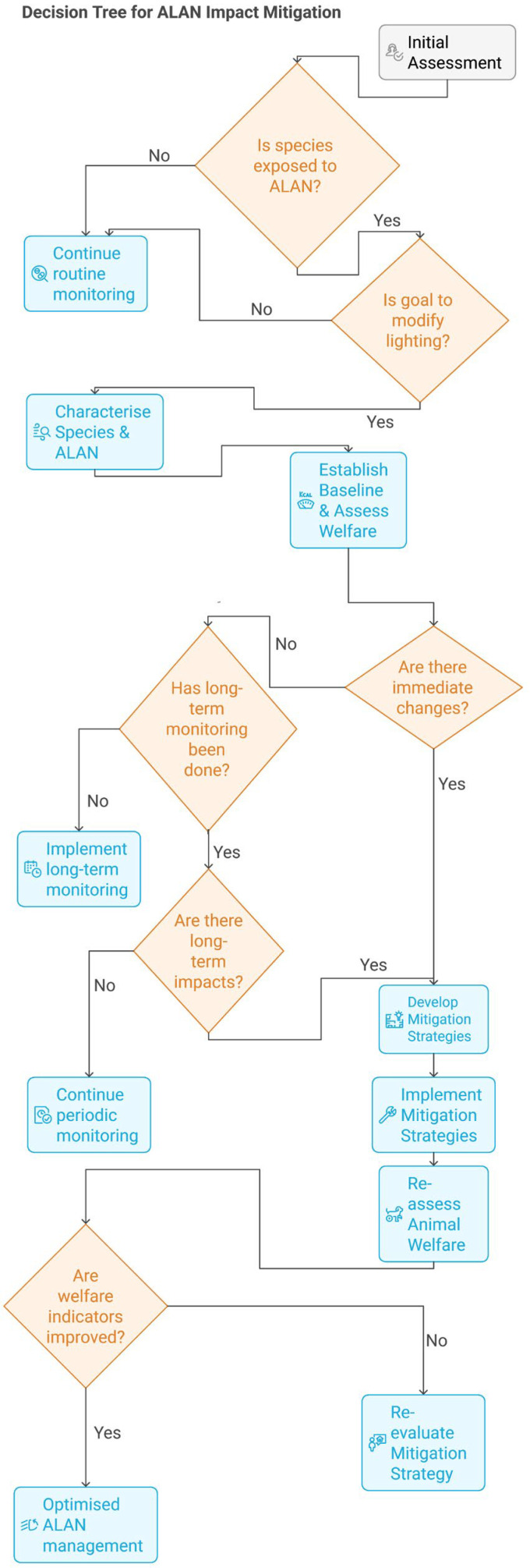
Decision tree illustrating a systematic approach to assess and mitigate the impacts of Artificial Light at Night (ALAN) on animal welfare across zoos and other animal-holding institutions. ALAN refers to artificial light sources that illuminate the environment during natural nocturnal periods, potentially disrupting animal behaviour and physiology (Authors’ own diagram).

**Table 1 animals-15-03329-t001:** Approximate critical flicker fusion frequencies (CFF) by taxa/groups and typical flicker frequencies of standard artificial lighting systems. Data are derived from Lafitte et al. (2022) [82]. This table provides mean ± standard deviation and observed range for each vertebrate group, highlighting the potential risk of visual stress under artificial lighting systems with frequencies below species-specific CFF thresholds.

Vertebrate Group	CFF (Hz) (Mean ± SD; Range)	Notes/Implications for Flicker Risk
Elasmobranchii	26.4 ± 11.7 (32–45)	Relatively low temporal resolution: most artificial lights are above the perceptual threshold.
(sharks, rays)		
Actinopterygii	49.9 ± 23.8 (8.8–117)	Wide variability; fast-swimming or diurnal species may detect flicker at lower frequencies.
(ray-finned fish)		
Amphibia	19 (19; one study)	Minimal data; likely low sensitivity to flicker.
Reptilia	45.9 ± 17.8 (21–80)	Moderate temporal resolution; some species may perceive flicker in older lighting systems.
Aves	86.7 ± 31.9 (28–143)	High temporal resolution; likely to perceive flicker from low-frequency lights (e.g., fluorescent lamps).
Mammalia	46.0 ± 18.1 (5–84)	Moderate resolution; flicker perception varies across species.

## Data Availability

No new data were created or analyzed in this study. Data sharing is not applicable to this article.

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
