# Peer review of "The Effects of Light on Vertebrate Welfare: A Review"

_animals, 2025, doi:10.3390/ani15223329_

Round 1
Reviewer 1 Report
Comments and Suggestions for Authors
I thank the authors for this very informative and interesting read. The paper addresses the complex role of light in animal welfare, particularly within zoological settings, and the topic itself represents a novel and valuable contribution to the field.
However, a large segment of the manuscript reads more like a didactic textbook chapter rather than a scholarly review, tending to summarize information rather than critically analyze it. For example, Figures 1 and 2 are interesting and well-presented, but their relevance to animal welfare and zoo management practices is not made clear. How do these physical principles of light relate to the lived experience and welfare of zoo animals, or to husbandry practices?
Lines 78–85: Although I am not an expert on the effects of light on animal welfare, a quick search on Google Scholar reveals an abundant number of studies on lighting effects in domesticated animals (e.g., poultry, cattle, and laboratory rodents). These could be referenced or discussed briefly, as they may provide informative parallels for readers and strengthen the empirical grounding of this review.
Section 2 (“The physical characteristics of light”) reads largely like a textbook summary. It would be more engaging for Animals readers if the discussion were more directly linked to animal welfare and captive management contexts. For instance, in Lines 137–150, the authors describe mechanisms of light perception and physiology but refrain from connecting this knowledge to zoo practice. This section would be strengthened by explicitly identifying what is already known, what remains uncertain, and why these knowledge gaps matter for zoo animal welfare and management.
Sections 4 and 5 are of particular interest to scholars working on animal welfare, but they appear rather late in the paper and are somewhat underdeveloped. The authors could consider adopting a more coherent framing throughout—showing, for example, how light affects each of the Five Domains (nutrition, health, environment, behaviour, and mental state)—to create stronger conceptual unity and guide readers toward applied welfare implications.
Line 634–639: Please provide a few concrete examples to illustrate the point. These could include specific cases where light management strategies were implemented successfully (or unsuccessfully) in zoo settings, or examples of how visitor behavior (e.g., camera flashes or mobile phone lights) directly affected nocturnal animal welfare. Such details would enhance the practical and applied relevance of the review.
Author Response
I thank the authors for this very informative and interesting read. The paper addresses the complex role of light in animal welfare, particularly within zoological settings, and the topic itself represents a novel and valuable contribution to the field.
Response: We sincerely thank the reviewer for the positive feedback and constructive suggestions. We greatly appreciate the time and effort dedicated to reviewing our manuscript and for recognising the relevance and contribution of our study to the field.
However, a large segment of the manuscript reads more like a didactic textbook chapter rather than a scholarly review, tending to summarize information rather than critically analyze it. For example, Figures 1 and 2 are interesting and well-presented, but their relevance to animal welfare and zoo management practices is not made clear. How do these physical principles of light relate to the lived experience and welfare of zoo animals, or to husbandry practices?
Response: We thank the reviewer for the valuable and constructive comments. Our intention with this section was to provide readers with a concise background on the fundamental physical properties of light, as this topic is often unfamiliar to professionals working in zoo animal welfare. We understand, however, that the section became overly theoretical and resembled a textbook summary. In response, we have substantially shortened this section, retaining only a brief explanation of the light spectrum (wavelengths) to support the discussion that follows. We also improved the connection between the physical aspects of light and their implications for animal welfare and husbandry practices in zoological contexts. Section 3, however, was retained with minor modifications, as it provides essential information on animals’ perceptual abilities, which, in turn, supports comprehension of the subsequent sections of the review.
Lines 78–85: Although I am not an expert on the effects of light on animal welfare, a quick search on Google Scholar reveals an abundant number of studies on lighting effects in domesticated animals (e.g., poultry, cattle, and laboratory rodents). These could be referenced or discussed briefly, as they may provide informative parallels for readers and strengthen the empirical grounding of this review.
Response: Thank you very much for your insightful comment. We have already cited references on domestic animals in our discussion and will expand the discussion further using these examples throughout the text. The new insertions are highlighted in red.
Section 2 (“The physical characteristics of light”) reads largely like a textbook summary. It would be more engaging for Animals readers if the discussion were more directly linked to animal welfare and captive management contexts. For instance, in Lines 137–150, the authors describe mechanisms of light perception and physiology but refrain from connecting this knowledge to zoo practice. This section would be strengthened by explicitly identifying what is already known, what remains uncertain, and why these knowledge gaps matter for zoo animal welfare and management.
Response: We thank the reviewer for this valuable comment. We agree that the previous version of Section 2 read more like a textbook summary and did not sufficiently connect the physical aspects of light to zoo animal welfare and management. We have now revised this section to focus more directly on the relevance of light properties to the perception, behaviour, and welfare of animals under human care. The new version better links physical principles, such as wavelength, reflection, and refraction, to practical implications for captive environments and husbandry design. The revised text is now highlighted in red in the manuscript.
Sections 4 and 5 are of particular interest to scholars working on animal welfare, but they appear rather late in the paper and are somewhat underdeveloped. The authors could consider adopting a more coherent framing throughout—showing, for example, how light affects each of the Five Domains (nutrition, health, environment, behaviour, and mental state)—to create stronger conceptual unity and guide readers toward applied welfare implications.
Response: Thank you very much for this valuable comment. We have revised Section 4 to highlight more clearly the influence of light on animal welfare using the Five Domains model (nutrition, health, environment, behaviour, and mental state). We believe that this restructuring has improved the conceptual coherence of the manuscript and enhanced its overall clarity.
Line 634–639: Please provide a few concrete examples to illustrate the point. These could include specific cases where light management strategies were implemented successfully (or unsuccessfully) in zoo settings, or examples of how visitor behavior (e.g., camera flashes or mobile phone lights) directly affected nocturnal animal welfare. Such details would enhance the practical and applied relevance of the review.
Response: Thank you very much for this helpful comment. We have expanded this section to include concrete examples of light management strategies in zoo contexts, as suggested. The revised text now incorporates documented cases of nocturnal exhibits for slow lorises, as well as broader examples of the impacts of artificial light on zoo-housed nocturnal species. These insertions, highlighted in red, strengthen the applied dimension of the review and demonstrate how evidence-based lighting design can enhance animal welfare and visitor management practices.
Reviewer 2 Report
Comments and Suggestions for Authors
Animals 3921180 Effects of light on zoo vertebrate welfare: a review
This is a valuable paper and excellent contribution. It is well constructed and takes the less informed reader clearly through the basic knowledge. I have one major point to raise to the authors, which is the basis for my few (and minor) suggestions.
My rationale:
Many millions of vertebrates (of wild species) are kept in captivity and for many species this is predominantly in the exotic pet market, whether we are talking slow loris, squirrels, otters, small rodents or small primates, let alone the birds, reptiles / amphibia and fish. These may be kept by ‘hobbyists’, breeders, sellers or someone wanting one or two individual animals as pets. Bottom line is that overall, your paper is relevant to all these groups. By highlighting only the zoo environment, I fear that you are doing a dis-service to your work and to animal welfare more broadly and this will also limit readership and citation of this important paper.
Hence my suggestions:
- Please remove the word Zoo from the title. The effects of light on vertebrate Welfare: a review
Other suggestions are tabled in the attached document. These are minor and mainly for clarity, or as above to show the width of applicability for this paper

Author Response
This is a valuable paper and excellent contribution. It is well constructed and takes the less informed reader clearly through the basic knowledge. I have one major point to raise to the authors, which is the basis for my few (and minor) suggestions.
My rationale:
Many millions of vertebrates (of wild species) are kept in captivity and for many species this is predominantly in the exotic pet market, whether we are talking slow loris, squirrels, otters, small rodents or small primates, let alone the birds, reptiles / amphibia and fish. These may be kept by ‘hobbyists’, breeders, sellers or someone wanting one or two individual animals as pets. Bottom line is that overall, your paper is relevant to all these groups. By highlighting only the zoo environment, I fear that you are doing a dis-service to your work and to animal welfare more broadly and this will also limit readership and citation of this important paper.
Response: We sincerely thank the reviewer for this thoughtful and encouraging feedback. We agree that the topic discussed in our paper extends beyond zoological institutions and is equally relevant to animals kept in other captive contexts, including domestic, production, and exotic pet settings. Following this valuable suggestion, we have revised the text to broaden the scope of the manuscript. The revised version now integrates examples from both zoo-housed species and domesticated or production animals to enhance the general applicability and relevance of our discussion. These additions are highlighted in red throughout the text.
Hence my suggestions:
- Please remove the word Zoo from the title. The effects of light on vertebrate Welfare: a review
Response: Removed, as suggested.
Other suggestions are tabled in the attached document (pasted below). These are minor and mainly for clarity, or as above to show the width of applicability for this paper.
Animals 3921180 Effects of light on zoo vertebrate welfare: a review
This is a valuable paper and excellent contribution. It is well constructed and takes the less informed reader clearly through the basic knowledge. I have one major point to raise to the authors, which is the basis for my few (and minor) suggestions.
Response: We are deeply grateful for your insightful comments and positive assessment of our manuscript. Your feedback, particularly regarding its construction and clarity for the less informed reader, is highly appreciated. We also thank you for raising your major point and for your valuable suggestions, which have undoubtedly enhanced the quality and comprehensiveness of our work.
Other suggestions
Response: In the table, accepted suggestions are highlighted in green, unaccepted suggestions are marked in red, and an appropriate response is provided in a new column. Within the article text, modifications are highlighted in red.
|
line |
from |
Change to |
Rationale |
Response |
|
23 |
In zoos, however, |
In captivity, however |
Narrow down to zoo concept as appropriate |
|
|
25 |
For example, animals kept in nocturnal houses |
For example, animals kept in zoo nocturnal houses |
|
|
|
31 |
but their use in zoos is still limited |
but their use is still limited |
|
|
|
38 |
yet its management in zoological settings remains, relatively, under-explored |
yet its management in captive settings remains, relatively, under-explored |
Patience for the zoo bit ? |
|
|
46 |
Zoo management practices, such as reversed |
With a focus on zoos, management practices, such as practices, such as reversed |
|
|
|
73 |
of its influence on the welfare of zoo animals is light. |
of its influence on the welfare of captive animals is light. |
I am trying keep the generis captive issue as well as the specific zoo issues in the reader’s mind. |
|
|
85 |
context of zoos, and more learned in the ecological context |
context of zoos, compare to the ecological context |
|
|
|
102 |
and artificial lighting from lamps are also light sources that can affect |
and artificial lighting are also sources that can affect |
Both ‘lamps’ and the repetition of ‘light source’ are redundant in this sentence. |
This sentence was removed from the text after the implementation of reviewer 1’s comments. |
|
109 - 111 |
their wavelength (measured in nanometers); animals perceive the visible part of this spectrum (380 to 750 nm) as different colours |
their wavelength, measured in nanometers. Humans see light in the range of 380 to 750 nm, known as the “visible” part of the spectrum, and animals perceive this part as |
Shorter sentences provide clarity here. As it currently reads there is a bit of confusion about whose visible spectrum it is .. given we know e.g. some mammals |
This sentence was changed after the implementation of reviewer 1's comments. |
